# Natural *Plasmodium falciparum* Infection Stimulates Human Antibodies to MSP1 Epitopes Identified in Mice Infection Models upon Non-Natural Modified Peptidomimetic Vaccination

**DOI:** 10.3390/molecules28062527

**Published:** 2023-03-10

**Authors:** Zully Johana Rodríguez, Fredy Leonardo Melo, Angela Torres, Nikhil Agrawal, Jesús Alfredo Cortés-Vecino, José Manuel Lozano

**Affiliations:** 1Departamento de Farmacia, Mimetismo Molecular de los Agentes Infecciosos, Universidad Nacional de Colombia-Sede Bogotá, Carrera 30#45-03, Bogotá DC 111321, Colombia; 2Departamento de Química, Universidad Nacional de Colombia-Sede Bogotá, Carrera 30#45-03, Bogotá DC 111321, Colombia; 3College of Health Sciences, Discipline of Pharmaceutical Sciences, University of KwaZulu-Natal, Westville, Durban 4000, South Africa; 4Laboratorio de Parasitología Veterinaria, Universidad Nacional de Colombia-Sede Bogotá, Carrera 30#45-03, Bogotá DC 111321, Colombia

**Keywords:** malaria vaccine, site-directed modification, synthetic epitopes, peptide-bond isostere

## Abstract

(1) Background: Malaria, a vector-borne infectious disease, is caused by parasites of the *Plasmodium* genus, responsible for increased extreme morbidity and mortality rates. Despite advances in approved vaccines, full protection has not yet been achieved upon vaccination, thus the development of more potent and safe immuno-stimulating agents for malaria prevention is a goal to be urgently accomplished. We have focused our research on a strategy to identify *Plasmodium* spp. epitopes by naturally acquired human antibodies and rodent malaria infection models immunized with site-directed non-natural antigens. (2) Methods: Some predictive algorithms and bioinformatics tools resembling different biological environments, such as phagosome-lysosome proteolytic degradation, affinity, and the high frequency of malaria-resistant and -sensitive HLA-II alleles were regarded for the proper selection of epitopes and potential testing. Each epitope’s binding profile to both host cells and HLA-II molecules was considered for such initial screening. (3) Results: Once selected, we define each epitope-peptide to be synthesized in terms of size and hydrophobicity, and introduced peptide-bond surrogates and non-natural amino acids in a site-directed fashion, and then they were produced by solid-phase peptide synthesis. Molecules were then tested by their antigenic and immunogenic properties compared to human sera from Colombian malaria-endemic areas. The antigenicity and protective capacity of each epitope-peptide in a rodent infection model were examined. The ability of vaccinated mice after being challenged with *P. berghei* ANKA and *P. yoelii* 17XL to control malaria led to the determination of an immune stimulation involving Th1 and Th1/Th2 mechanisms. In silico molecular dynamics and modeling provided some interactions insights, leading to possible explanations for protection due to immunization. (4) Conclusions: We have found evidence for proposing MSP1-modified epitopes to be considered as neutralizing antibody stimulators that are useful as probes for the detection of *Plasmodium* parasites, as well as for sub-unit components of a site-directed designed malaria vaccine candidate.

## 1. Introduction

Currently, the number of clinical cases and death registered for malaria [1] show that there remains a need for progress towards new knowledge about molecular mechanisms for the immune response against infection of *Plasmodium* spp. Numerous vaccine candidates against the disease have been developed, targeting complete, irradiated, and/or attenuated pre-erythrocytic forms of the parasite, to recombinant antigens and synthetic peptides, formulated in a variety of adjuvants and technology platforms [2]. However, all these interventions have limited efficacy [3].

The most representative advance in this field is the vaccine called RTS, S/AS01E, directed against pre-erythrocytic stage antigens, specifically the CSP protein, which was recently approved by the WHO for use in African children [4]. A significant decrease (30%) in severe and fatal malaria cases has been observed. The IgG induced by the vaccination is significantly related to its protective activity [5]. The effect of a booster vaccination with a fractional dose of this vaccine in a controlled human malaria infection context has been studied, showing that boosters given one year after completion of a primary two or three doses was well tolerated and can extend or induce protection to control human malaria infection [6].

Recent efforts have focused on the identification of antigens from merozoites and sporozoite surface proteins implicated in cell invasion [7]. The merozoite surface proteins (MSPs) are one of the most important families of antigens of *Plasmodium* spp. [8,9]. In particular, MSP1 has established itself as an interesting target. It is a 195 kDa protein, synthesized in schizogony, with conserved, semi-conserved, and variable sequences [10]. Its proteolytic degradation fragments are important in red blood cell invasion processes. Antibodies against this protein could interrupt infection, which has been demonstrated in the Aotus monkey model [11], in which they used a candidate of recombinant origin, which initially showed a significant difference in the prepatent period after challenge with sporozoites, but in a second study, this was debated [12].

Those native proteins have exhibited potential as immunogens and have been step-by-step disclosing novel structural elements that increase the knowledge of *Plasmodium* spp. pathogenicity, therefore our understanding to propose specifically modified epitope-peptide fragments as potential components of a new generation of multicomponent malarial vaccine candidates. This practical perspective exhibits some considerable disadvantages, mainly related to the reduction in bioavailability by the degradation of proteolytic in vivo and the low immunogenicity of native sequences [13]. This immunological profile can be changed by the incorporation of specific conformational modifications or structural modifications that are site-directed by a single replacement of a peptide bond on the antigen backbone, which can influence the molecular structure of the proposed epitope while keeping the genetic information of the pathogen. We used the term peptidomimetics as a nomenclature adapted from [14] to refer to structurally modified peptides.

Peptidomimetics can stimulate cellular and humoral immune responses to neutralize infection and modulate the recognition of the vertebrate host immune system, thus achieving the conversion of non-relevant peptides at the immunological level into structures with immunogenic potential [15].

This research aimed to evaluate, for the first time, both Fmoc chemistry strategies for obtaining the native and peptide-bond isostere-modified sequences of each selected epitope-peptide based on the *Plasmodium* spp. MSP-1 antigen, and their immunological effect in stimulating mice anti-peptide antibodies capable of controlling the provoked disease in two experimental rodent models. The assessment of anti-peptide antibodies’ cross-reactive properties regarding those of naturally acquired human antibodies from endemic malaria areas of Colombia is a relevant matter. Obtained results highlight the potential application of modified immune potentiating probes as a possible component of a vaccine against malaria.

How specific epitope-peptide sequences are selected, processed, and presented by the human leukocyte antigen (HLA) system of an individual, as well as the clues for their influence on the cellular immune response, remain a knowledge gap. Some bioinformatics tools approaches have provided a variety of ideas regarding a number of potential B- and T-epitopes. Interestingly, isolated lymphocytes from selected vaccinated mice in ex vivo assays allowed us to identify Th-cell patterns in response to vaccination with modified specific MSP1 sequences when used for stimulating antibodies against *Plasmodium* spp.

## 2. Results

### 2.1. Bioinformatic Analysis and MSP1-Epitope-Peptide Selection

The epitope-peptide selection included a multi-step strategy as further described, starting from genomic analysis of human and rodent *Plasmodium* spp. MSP1 gene products. Therefore, amino acid sequence identity and protein structure homology for the MSP1 antigen was compared among available human and rodent *Plasmodium* species, being *P. falciparum* NF54 (W7KFP9_PLAFO), *P. falciparum* Wellcome (P04933), *P. falciparum* FVO (A0A024V850_PLAFA), *P. falciparum* FCR3 (A5A7B1_PLAFA), *P. falciparum* FC27 (P13819), and the reference *P. falciparum* 3D7 strain for a protein of 1720 amino acids (PF3D7_0930300), in comparison with MSP1 gene products from *P. berghei* ANKA (PBANKA_0831000) and *P. yoelii* 17XL (PY17X_0834400) of 1791 and 1772 amino acids, respectively. The MSP1 antigen’s primary structure from the human reference strain *P. falciparum* 3D7 served as the basis for a comparison regarding the gene products from rodent *Plasmodium* strains (Figure 1A). Following the proposed steps for selecting the base sequences of this research as described in Materials and Methods lead us to identify four **B** and one **T** epitopes on the MSP1 structure. From these analyses, the location of both *N*- and *C*-termini MSP1 representative target fragments can be observed in Figure 1A, in which identified B epitopes were ^38^AVLTGYSLFQKEKMVLNEGTS^58^, ^42^GYSLFQKEKMVLNEGTSGTA^61^, ^1545^YLKPLAGVYRSLKKQ^1560^, and ^1803^MLNISQHQCVKKQCPQNSY^1821^, coded as **B1**, **B1.1**, **B6**, and **B7**, respectively, and the coded **T3** for a T-epitope ^217^QIPFNLKIRANELDVLKKLV^236^. A significant homology percentage was estimated based on percent similarity and identity. Identity score averages for all five epitopes ranged from 28.6% to 45.0% among the three compared MSP1 *Plasmodium* spp. sequences while having similarity average values from 57.1% to 75.0% (Figure 1B).

The next experimental step consisted of synthesizing all five MSP1 epitope sequences and a family of non-natural peptide analogs among **-ψ-[CH_2_–NH]-** peptide isostere bonds (backbone modifications) and single ***D*-amino acid** substitutions (side-chains chirality) by liquid- and solid-phase Fmoc chemistry and then spectrophotometric characterization. Each peptide surrogate was strategically positioned as designed (Table 1).

### 2.2. Functional Assessment of MSP1 Native, -ψ-[CH_2_–NH]-Peptide-Bond Isosteres and D-Amino Acid Modified Sequences

Anti-peptide-epitope antibody titers and their reactivity to MSP1 were determined in mice sera by a standardized ELISA method as described. Epitope-peptides and their non-natural analogs were formulated with Freund’s adjuvant and i.p. administered in a three-dose immunization scheme. Once antibody titers showed mid–high reactivities, each animal group was split and submitted to an experimental challenge by i.p injection of 5 × 10^4^ infected red blood cells (iRBCs) per animal with *P. berghei* ANKA and *P. yoelii* 17XL strains in simultaneous experiments. Parasitemia levels of all MSP1-epitope-vaccinated animals were followed up to the 20th day post-challenge. Figure 2A–E show the parasitemia control profile for both parasite-challenged groups, survival percentage, antigenicity, and antibody titers, highlighting those animals that showed significant statistical differences concerning the control (pre-immune serum), as evidence of antigenic recognition as seen in Table 1.

In agreement with the molecular design observed in Table 1, all proposed designed B- and T-epitopes were obtained, formulated, and employed for i.p. mice vaccination in a four-dose scheme. Experiments included a potential T epitope in vaccination. Obtained results led us to a better understanding of the isostere peptide bond’s role in modulating a molecular structure by introducing possible B-epitope molecular features, as evidenced by stimulated humoral response upon vaccination. Polyclonal antibodies in blood samples taken a day before the animal groups were split and challenged with two different rodent malaria species were tested. As a consequence, the assessment of each designed epitope scope against the disease was fulfilled. Anti-peptide antibody titers were determined with samples after the third boost by standard ELISA assays as mentioned in Materials and Methods. As seen in Table 1, antibody titers stimulated by native sequences were in a low–mid range, while those isostere-peptide-bond-containing molecules exhibited higher antibody titers in some remarkable cases.

Follow-up of vaccine-challenged mice on the severity of the disease caused by *Plasmodium* spp. showed that animals of the control groups normally died within the first 7 days after being challenged in agreement with previous observations [16]. On the other hand, some groups of animals immunized with MSP1 peptidomimetics revealed survival profiles between 25% and 50% (Figure 3).

The survival profile of these animal groups evidenced differential immunotherapeutic effects of the designed epitope-peptides depending on their amino acid composition or the presence of non-natural elements introduced on their structure, eliciting not only an antibody response but a protective effect against the lethal challenge with both rodent *Plasmodium* strains.

As shown in Figure 2 and Figure 3, parasitemia levels of tested mice revealed that some sequences seem to stimulate functional neutralizing antibodies which exhibit strong immunoreactivity against the two strains of human *P. falciparum*. Anti-peptide antibody reactivity for *P. falciparum* surface proteins showed specific reactivities revealing bands having relative electrophoretic mobility at 33, 42, 83, 150 to 200 kDa, and 83 kDa, especially those elicited by **B1** and **B1.1** peptide-epitope analogs derived from the MSP1 *N*-terminal, as well as characteristic bands revealed by anti-**B6**, **B7,** and **T3** analogs’ antibodies representative of the MSP1 surface antigen, therefore suggesting that selected peptide-epitopes could be regarded as potential targets to stimulate protective immunity against malaria as discussed later.

The **B1** and **B1.1** overlapping sequences have been previously studied for their immunogenic capacity to stimulate human T lymphocytes, as well as because both harbor high-affinity binding motifs to RBCs [17,18,19]. Previously, the MSP**1**-Y^43^-ψ[CH_2_NH]-S^44^ peptide-bond isostere, herein coded as **B1An2,** has been described as a determinant for recognition by T lymphocytes in the context of human-HLA-II-defined alleles [13]. Thus, this potential immunodominant epitope has evidenced in our studies the ability to modulate a given immune response as tested in the animal model since vaccinated animals efficiently controlled *Plasmodium* parasitemia levels and some of them survived the lethal experimental challenge.

Interestingly, the group of animals immunized with the **B1An4** (M^51^-ψ[CH_2_-NH]-V^52^) modified epitope suggests that by correctly positioning a given-ψ[CH_2_–NH] peptide-bond isostere, a protective response could be elicited in the vaccinated animals. In contrast, the modification on MSP1-G^42^-ψ[CH_2_NH]-Y^43^ of the **B1An1** analog does not modulate a comparable response and therefore does not present a protective profile, suggesting that it does not favor conformational changes that allow a desired immunological activity. Similarly, with the low protective activity stimulated by the MSP1-L^46^-ψ[CH_2_NH]-F^47^ modification of **B1An3**, a non-significative antibody stimulation was induced. As observed in this study, the **B1.1** peptide family members did not present significant results in terms of stimulation of an efficient protective effect of vaccinated animals when challenged with rodent *Plasmodium* strains. The (M^51^-ψ[CH_2_NH]-V^52^) modification present in the **B1** epitope-peptide analog lacks its protective stimulating effect when is present in the **B1.1**-member-coded **B1.1An2,** whose *N*-terminus **has a** larger sequence and seems to play a role in MSP1 recognition being a key piece of the smaller **B1**-modified peptide-epitope for stimulation of monoclonal antibodies [20].

On the other hand, the introduction of side-chain chiral substitutions with non-natural D-amino acid replacements on the ^1286^LeuLysPro^1288^ motif from the MSP1-*C*-terminal peptide **B6,** associated with critical residues of binding to red blood cells [19], stimulates the production of antibodies that favor protection against the experimental challenge, as can be seen in **B6An1** (-Y-*d*L^1546^-K-) and **B6An2** (-L-*d*K^1547^-P-), efficiently controlling the parasitemia levels in immunized animals. As demonstrated in previous immunization experiments of Aotus monkeys with these two peptide members obtained by using a formerly used solid-phase strategy named *t*-boc (tert-butyloxycarbonyl *N-*protected amino acids), anti-modified peptide-stimulated antibodies reacted with surface proteins of *Plasmodium falciparum* FCB2 membrane lysates [21]. The current work evidenced that the immunological effect seems to be mainly affected if the chiral modification is carried out on the amino acid proline of the **B6-An3** (-K-*d*P^1548^-L-) analog.

The *N*-terminal end of MSP1, where the epitope-peptide named **B7** is located, specifically in the 19 kDa fragment of MSP1, has been involved in the formation of *Plasmodium* parasitophorous vacuoles, and it has been shown that immunization of *Aotus* monkeys with this peptide sequence generated the production of antibodies seems to induce immunogenic protection by antibodies which also reacted with proteolytic fragments of MSP1 [22]. Remarkably, from the current work, peptide backbone modification of amino acid residues located upstream from the RBC binding motif of this peptide fragment transforms this sequence into a protective epitope represented by the **B7An2** (L^1804^-ψ[CH_2_-NH]-N^1805^) peptide analog.

Concerning the MSP1-83kD **T3** (^217^QIPFNLKIRANELDVLKKLV^236^) epitope-peptide, at least three mice from the vaccinated group efficiently controlled the malaria infection overcoming the follow-up time, specifically mice immunized with **T3An4** (N^227^-ψ[CH_2_-NH]-E^228^) and **T3An8** (L^232^-ψ[CH_2_-NH]-K^233^). The other analogs of the **T3** family showed similar behavior to the control group, with an accelerated increasing infection in the first days and the death of all animals occurring between day 7 and day 11 post-infection. Evidence suggests that the peptide backbone and stereochemical modifications seem to play an important role in developing novel molecular tools to prevent infectious diseases [21,23,24].

### 2.3. Individuals Exposed to Natural Plasmodium spp. Infection

In Colombia, 67.0% of the total cases of malaria come from the Pacific region, with the department of Chocó reporting the most cases of malaria in the country, followed by the Nariño (20.6%) and Córdoba (11.4%) departments. The observed Annual Parasitic Index (API) for 2019 was 10.01 malaria cases per 1000 inhabitants at risk. The Chocó, Nariño, and Córdoba departments (provinces) are zones having high (API) values between 28.4 and 17.2 in the same year, while the Bolivar department displayed a low API while Bogotá DC does not have an API value. This means that the selected areas for this study are considered the zones of the country with historical endemicity, given the high prevalence of parasite-caused infections [25]. Based on the above, four malaria-endemic areas in Colombia were selected for collecting human sera samples (Figure 4A). Western blot analyses that aimed to asses human antibody reactivity were conducted with *Plasmodium falciparum* 3D7 strain, and showed that 100% of samples from human volunteers from Tumaco (Nariño department), 2% of samples from San Juan de Nepomuceno (Bolívar), 87% from Quibdó (Chocó), and 79% from Tierralta (Córdoba) strongly recognize those characteristic MSP1 protein bands of an SDS-PAGE resolved surface protein lysate while no recognition was shown by human sera samples from Bogotá DC, as expected. Western blot analysis regarding surface protein lysates from the *P. falciparum* FCB-2 revealed a similar pattern as above with strong reactivities for 75% of samples from human volunteers from Tumaco (Nariño), 8% of samples from San Juan de Nepomuceno (Bolívar), 75% from Quibdó (Chocó), and 78% from Tierralta (Córdoba), and no recognition bands of sera samples from Bogota DC were detected.

Antigenicity studies conducted by standardized (enzyme-linked immunosorbent assay) ELISA tests displayed that the five proposed epitopes selected as promising were strongly recognized as exhibiting high optical density (OD) values in an important number of sera samples of human volunteers from malaria-endemic areas as shown in Figure 4B, in contrast to control peptide antigens which had no reactivity for any sera sample. In most cases, the recognition percentages ratio was higher than 50% (Figure 5). Previous studies have demonstrated the role of immunoglobulin G (IgG) sub-classes in protection against the natural malaria disease caused by *P. falciparum* infection [26,27]. The immunoglobulin isotype repertoire in the analyzed human sera samples conforms to those antibodies stimulated by natural malarial infection episodes and specifically reacts with the artificially produced epitope-peptides as designed in this study.

Antibody recognition profiles of human sera samples from villages and cities named Tierralta (Córdoba department), Quibdo (Chocó department), and Tumaco (Nariño department), places of high malaria transmission, exhibited differential recognition patterns for most of the designed **-ψ[CH_2_-NH]**- peptide bond isosteres as observed in Western blot experiments. A remarkably strong reactivity of human antibodies for slow-mobility protein bands between 150 and 250 kDa seems to conform to a common recognition pattern among all samples from high-transmission malaria areas. A second, faster band up to 150 kDa, and a third band having mobility between 100 kDa and 150 kDa, seem to also be characteristic, as well as a strong band observed between 75 kDa and 100 kDa (Figure 5A–C). Interestingly, a high correlation of *Plasmodium* surface protein recognition patterns by human sera antibodies from high transmission malaria areas, with the protein band recognition patterns of mice serum antibodies, especially those vaccinated with specifically designed MSP1 epitope-peptides and their analogs, becomes evident as seen in Figure 5E, since a similar protein band recognition revealed strong reactivity for bands between 150 kDa and 250 kDa, 100 kDa and 150 kDa, another from 75 kDa to 100 kDa, and a band close to 50 kDa, respectively (Figure 5F). On the other hand, the observed low reactivity of sera samples from the San Juan de Nepomuceno village (Bolivar department) evidenced a straight correlation with the known low malaria endemicity of this zone of the country, as shown in Figure 5D, and the absence of reactive antibodies from human samples from a non-endemic malaria area (Figure 5E).

### 2.4. The Cellular Th1/Th2 Immune Response of Protected Vaccinated Mice against a Malaria Experimental Challenge

It has been previously described that during malaria infection episodes, differentiated blood stages of *Plasmodium* spp. stimulate a balanced innate immune response between pro-inflammatory (TNF-α, IFN-γ, and IL-2) and anti-inflammatory (IL-4 and IL-5) cytokines produced by polarized Th1 and Th2 lymphocytes, respectively, suggesting that cell-mediated immunity is under tight control to achieve immunity and to avoid a severe malarial pathology [28,29].

To determine cytokine stimulation on cell culture supernatants from spleen cells containing LB-lymphocytes derived from those mice that survived and resolved the experimental malaria challenge, an ex vivo cell culture experiment was performed in which spleen cells were three-times pulsed with 200 nM at times 0, 48, and 72 h with the corresponding epitope-peptide molecule used for vaccination (Table 2). Overall results suggested that some specific epitope-peptide analogs predominantly stimulate a profile of **Th1**-polarized lymphocytes by mainly eliciting high levels of **IL-2** and **IL-5** (in a smaller proportion), while TNF-α and IFN-γ were lesser stimulated, revealing a significant difference regarding the control group expressing cytokines (spleen cells from non-treated mice) as will be subsequently discussed in some detail. This fact could be directly related to macrophage activation, a pro-inflammatory response to vaccination in a cell-mediated immunity, where B lymphocyte differentiation and antibody secretion are promoted as an adaptative immunity mechanism. In contrast, in a natural acute infection, a pro-inflammatory response, with a high expression of INF alpha and gamma, stimulates macrophage activation, and the *Plasmodium* parasite itself modulates the expression of IL-10 and redirects an immune Th2 response, abolishing the malaria-protective Th1 mechanism as previously demonstrated [30,31]. As elsewhere described, *Plasmodium* spp. erythrocytic stages trigger a strong IFN-γ expression during acute infections of mice malaria models with *P. berghei*, *P. yoelii*, and *P. chabaudi*, as well as with *P. falciparum* in human malaria [28].

On the other hand, when designing a malaria subunit vaccine candidate, a specific molecular recognition of antigens and their binding to MHC-II which activate T-helper cells for subsequent adaptative immunity must be considered as an important task. For this reason, we have selected some HLA class II alleles that can be associated with sensibility and resistance to *Plasmodium* spp. infection to analyze the most effective designed epitope-peptide immunogen-binding profiles, aiming to correlate their functional activity and potential role in controlling the experimental disease. Representative class-I and II HLA alleles associated with sensitivity malaria are HLA-A*30:01, HLA-A*33:01, HLA-B*53:01, HLA-Cw*04:01, and HLA-DRB1*08:04, HLA-DRB1*13:02, and some alleles related to malaria resistance are HLA-B*53:01, HLA-DRB1*01:01, HLA-DQB1*05:01, HLA-DRB1*13:02, HLA-DRB1*13:01, HLA-DRB3*03:01, and HLA-DQA1*01:02, among others [32,33]. Therefore, molecular docking experiments would reveal differential interactions toward the immunodominance of these epitope-peptides for a proper malarial vaccine component selection.

### 2.5. In Silico Studies for the MSP1^42–61^ Fragment, Its Modified Analogs, and Their HLA-Peptide-TCR Complexes

To analyze possible differential intermolecular interactions between three-molecular complexes, consisting of either a malaria-resistant or -sensitive HLA-II presenting an anti-infection epitope-peptide to a TCR molecule, led to series of hydrogen bonding, salt bridges, and amino acids interactions being recorded, which were subsequently compared in stable complex states. Thus, HLA-DRB*101:01 (PDB code 1aqd) and HLA-DRB1*04:01 (1q94) were alleles for malaria resistance and susceptibility, respectively [34]. Coordinates for TCR-hemagglutinin peptide-HLA-II complexes were employed (1j8h) [35]. The MSP1^42–61^ fragment (2mu7) and coordinate pdb files for the two peptidomimetic analogs located at MSP1-V^52^ψ[CH_2_-NH]-L^53^- (**B1.1 analog 1**) and MSP1-M^51^ψ[CH_2_-NH]-V^52^- (**B1.1 analog 2**) were resolved by solution NMR [15,36].

As can be observed in Appendix A and Figure 6A–C, the MSP1^42–61^ native fragment established three hydrogen bonds between its Thr57 and Ser58 residues with the Gln9, Ans62, and Gln9 of the alpha chain of the malaria-sensitive HLA-DRB10401, while establishing two hydrogen bonds between its Gly42 and Phe46 with Asn69 and Asn62 of the alpha chain of the HLA-DRB10101. Regarding its interaction with both HLA-II beta chains, three hydrogen bonds between Gly42, Lys48, Ser58, Ala61, and two Asn54, with beta chain residues Gln62, Gln70, Asp28, and Asn82, as well as a saline bridge between MSP1 Lys48 with the Asp28 of this HLA-II beta chain, were observed. Therefore, hydrogen bonds and salt bridge interaction patterns strongly differ between these two systems. On the other hand, only the malaria-resistant HLA-II alleles establish a saline bridge between the epitope-peptide Lys48 residue with the Glu94 of the TCR alpha chain, while no interactions were observed when presented by the sensitive HLA-II allele.

Correspondingly, interactions of the MSP1-V^52^ψ[CH_2_-NH]-L^53^- (**B1.1 analog 1**) with the malaria-sensitive HLA-DRB10401 and the malaria-resistant HLA-DRB10101 alpha and beta chains regarding their interactions with TCR residues were analyzed. Hence, four hydrogen bonds were established first between Gln47 with Glu55, two hydrogen bonds between Thr57 with Asn69, and the other between the Ser58 with Gln64 of the sensitive HLA-II alpha chain.

Regarding this HLA-II beta chain interaction with this analog, one hydrogen bond between Asn54 with the chain Gln70 and a salt bridge between Glu55 with Lys71 were detected. In contrast, in this analog interaction with the malaria-resistant HLA-II, two hydrogen bonds were established between Lys48 and Glu49 with Asn69 and Glu11 of this HLA allele alpha chain, as well as a salt bridge between this analog Lys48 with the alpha chain Glu11. A hydrogen bond was established between Thr57 and the Gln64 of the malaria-resistant HLA-II allele. Remarkably, one hydrogen bond between the modified epitope Asn54 and the Asp51 of the TCR beta chain in the context of the sensitive HLA-allele was established, while a π-π (aromatic interaction) was revealed between the Try43 of this analog with the Tyr31 of the alpha chain of the TCR in the context of malaria-resistant HLA alleles.

Lastly, interactions between the MSP1-M^51^ψ[CH_2_-NH]-V^52^- (**B1.1 analog 2**) and the malaria-sensitive and -resistant HLA-II alleles, and later with the TCR residues, were analyzed. Gln47 analog residue forms a hydrogen bond with Asn69 of the sensitive HLA-alpha chain. Regarding this analog interaction with the sensitive HLA-allele beta chain, Gly42 formed two salt bridges with Asn57 and Glu9 of this chain, as well as between Lys50 and Gln70, Met51 and Gln70, Val52 and Gln70, and Gly56 and Asn82; Thr57 forms two hydrogen bonds with Asn82 and Gln9 of this allele and this complex interacts with the TCR alpha chain through the antigen Ser44 with the Asn99 and establishes a salt bridge between Lys48 and Glu102. Regarding interactions of this analog complexed with the malaria-resistant HLA-II allele, its Gln47 residue forms a hydrogen bond with Asn62, and interestingly the Lys 48 stabilizes three hydrogen bonds, with three different residues of this allele alpha chain being Asn62, Gln9, and Glu11, and one hydrogen bond between the Phe46 and Gln70 of this HLA-II allele beta chain. No molecular interactions of this complex were observed with the TCR.

### 2.6. RMSD, Radius of Gyration (RG), Root Mean Square Fluctuations (RMSF), Number of H Bonds, Solvent-Accessible Surface Area (SASA), Secondary Structure Changes in Peptides

Bearing in mind the relevance of the immunological properties of MSP1-V^52^ψ[CH_2_-NH]-L^53^- (**B1.1 analog 1**) and MSP1-M^51^ψ[CH_2_-NH]-V^52^- (**B1.1** analog 2) regarding experimental malaria infection control when used for vaccination, we decided to analyze their molecular structural stability as single molecules, without interacting with either an antigen-presenting class-II HLA or TCR molecules, by in silico molecular dynamic simulation studies. To investigate the overall conformational change in peptides, we have performed RMSD analysis for each peptide and time evolution of RMSD, Appendix A, and line plots show that all simulations were equilibrated. RMSD boxplot data show that the native peptide has high RMSD changes in both simulations compared to **B1.1 analog 1** and **B1.1 analog 2** simulations; **B1.1 analog 1** SIM1 shows the slowest RMSD changes. Overall, both the mutated peptide simulations show less conformational changes compared to the native peptide simulations. RG shows the compactness of the protein or peptide structure. Appendix A shows the time evolution and boxplots of RG. Time evolution data revealed that RG values for peptides fluctuated within a range of ~6 Å to ~14 Å, and overall RG data follow the same trends as RMSD and show that mutated peptides are more compact compared to the native peptide. RMSF data show (Appendix A) *N*-terminal residues and the central region residues have high fluctuations in both simulations in the native MSP1^42–61^ peptide compared to *C*-terminal residues. Additionally, in mutated peptides, *N*-terminal and *C*-terminal region residues show high fluctuations compared to the central region in both simulations of each mutated peptide. Time evolution of intra-peptide H bonds shows that there is a greater number of intra-peptide bonds formed in mutated peptides compared to the native peptide, and the native peptide shows more SASA (Appendix A) compared to mutated epitope-peptides. Time evolution of secondary structure change (Appendix A) reveals that the peptide contains more structure content in both simulations compared to the mutated peptide and in SIM2 of the native peptide we also observe that it obtains a β-sheet structure (Appendix A). Mutated peptides show less secondary structure content compared to the native peptide and most of the residues obtain a coil structure. However, we observed some α-helix content in the N-terminal region of mutated peptides for different periods.

## 3. Discussion

A site-directed peptide-based vaccine design led us to obtain a series of **B** and **T** potential epitope-peptides which were synthesized and immunologically characterized by in vivo and ex vivo experiments. Thus, the so-named **B1**, **B1.1**, **B6**, **B7**, and **T3** representative epitopes from the **MSP1** *Plasmodium* spp. structure were inoculated in groups of BALB/c mice to stimulate anti-peptide antibody titers, as proven by ELISA tests. Subsequent malaria infection was performed by injecting lethal doses of two rodent malaria strains (*P. berghei* ANKA and *P. yoelii* 17XL) for assessment of each immunogen’s capacity to control parasitemia levels over 20 days post-infection. Remarkably, vaccinated mice with some strategically positioned peptide bond isosteres and some chiral mutations on given immunogens were effective in handling infection, and some resolved the induced disease, which did not occur in animals from the control groups. Interestingly, those murine anti-peptide antibodies react similarly to human antibodies from malaria-endemic areas in Colombia, as shown by comparable protein bands revealed by Western blot experiments with *Plasmodium falciparum* 3D7 and FCB-2 strains. As observed in Western blot experiments, the most representative bands were those whose mobility was located at relative molecular weight ranges between 150 kDa and 250 kDa, 100 kDa and 150 kDa, 75 kDa and 100 kDa, and about 50 kDa. Human antibody recognition revealed this commonly seen banding pattern of *P. falciparum* proteins in membrane lysates upon being resolved in SDS-PAGE systems under reducing conditions and then being electro-transferred to nitrocellulose supports. Such banding patterns resemble both lysates of 3D7 and FCB-2 *Plasmodium* strains. Interestingly, the MSP1 *Plasmodium* surface antigen whose molecular weight ranges between 195 kDa and 200 kDa undergoes two proteolytic processes during parasite infection in RBCs. The first degradation step releases protein fragments of an *N*-terminus fragment of 83 kDa, and other fragments of 28–30 kDa, 38 kDa, and a *C-*terminus of 42 kDa which remains attached to the parasite membrane. Then, a second processing releases a 33 kDa protein fragment from the 42 kDa and a 19 kDa fragment remains attached to the parasite surface through a GPI anchoring motif. Therefore, evidence supports the hypothesis that bands from the entire MSP1 antigen at 195–200 kDa and its processed fragments at 83 kDa are specifically recognized. Moreover, a band of about 120 kDa, probably representing the MSP1 antigen after losing its *N*-terminal 83 kDa fragment, also has a strong reactivity with polyclonal antibodies of humans from high-transmission malaria zones. This antibody recognition pattern resembles that shown by the vaccinated mice depending on the location in the MSP1 structure of those selected epitope-peptides used for immunization.

On the other hand, ex vivo experiments conducted to determine a **Th1/Th2** cytokine expression pattern upon vaccination with the proposed MSP1-selected epitope-peptides were performed by mice spleen cell stimulation to determine cytokine expression by flow cytometry. Hence, an evident **Th1** cytokine pattern was detected on LB mice cells from animals vaccinated with **B1** (MSP1^38–61^), **B1** analog 2 (MSP1-M^51^ψ[CH_2_-NH]-V^52^-), **B1.1** (MSP1^42–61^), **B6** analogs 1 and 2 (MSP-1D-^1546^ and MSP-1^D-1547^), and **T3** (MSP-^1217–236^); therefore, the proposed molecular design proved to be in line with a subunit vaccine component selection in which isostere peptide bonds and chiral modifications seem to play a relevant role in a protective immune stimulation. On the other hand, **B1** analog 4 (MSP1-E^49^-ψ[CH_2_-NH]-K^50^) and **T3** analog 5 (MSP1-L^229^-ψ[CH_2_-NH]-D^230^) showed a balanced **Th1/Th2** profile in which both innate and adaptative immune mechanisms govern the immune response with regards to immunogen stimulation after vaccination and experimental challenge.

The 3D structures of some malaria-susceptibility and resistance-associated HLA class-II alleles were used for molecular dynamics and docking in in silico studies regarding the presentation of the most representative MSP1-designed epitope-peptides of this study in the TCR. Hence, when comparing molecular interactions between **B1** MSP1^42–61^ and its analogs MSP1-V^52^ψ[CH_2_-NH]-L^53^- (**B1.1** analog 1) and MSP1-M^51^ψ[CH_2_-NH]-V^52^- (**B1.1** analog 2), data displayed differential hydrogen bonding and salt bridge patterns for stable structure conformations for the three peptide immunogen molecules when interacting with the TCR. Some explanations and arguments regarding possible correlations between biological and in silico data could be inferred from these analyses.

Evidence presented in this study leads us to propose the application of similar strategies for designing and applying site-directed modified epitope-peptides derived from pathogen-representative antigens, aimed to stimulate efficient human antibodies harboring functional activity for infectious disease control and prevention. Altogether, these results will encourage and contribute to our understanding of the dynamics of the immune response after immunization with non-biological vaccines, which will be key for the development of strategies to control transmissible infections. Future studies considering ex vivo experiments in which human lymphocytes that consider the specific HLA restriction are selected could be conducted as an approach to elucidate the influence of certain peptide backbone modifications on the selected epitopes and therefore advance knowledge on the human immune response to those proposed peptidomimetics to identify them as potential components of sub-unit vaccines against *Plasmodium* spp. Blood stages.

## 4. Materials and Methods

### 4.1. Predicting MSP1 Epitopes, Bioinformatic Selection

Merozoite Surface Protein (MSP1) from the 3D7 reference *P. falciparum* strain [37] was selected for bioinformatics analysis and compared with homolog sequences from some human *Plasmodium* spp., such as *P. falciparum* FCB2, *P. falciparum* FCR3. *P. falciparum* FVO, *P. falciparum*. FC27, *P. falciparum* Wellcome, and *P. falciparum*. NF54, as well as regarding its identified orthologous sequences of *P. berghei* ANKA and *P. yoelii* 17XL, all of them reported in PlasmoDB http://plasmodb.org/plasmo/ (accessed on 8 December 2018) Protein Data Bank http://www.rcsb.org/pdb (accessed on 8 December 2018, and 16 December 2022) and Genbank NCBI http://www.ncbi.nlm.nih.gov databases (accessed on 8 December 2018, and 16 December 2022). Accession numbers (PF3D7_0930300.1), (PBANKA_0831000.1), and (PY17X_0834400.1), respectively. All MSP-1 sequences in FASTA format were submitted to multiple sequence alignment analyses by BLAST http://blast.ncbi.nlm.Nih.gov/Blast.cgi (accessed on 8 December 2018) and/or using the NPS@ (Network Protein Sequence Analysis) tool ClustalW available at https://npsa.lyon.inserm.fr/cgi-bin/npsa_automat.pl?page=/NPSA/npsa_clustalw.html (accessed on 8 December 2018), to establish the homology and identity degrees between the different MSP-1 sequences of *Plasmodium* spp. Analytical results lead us to propose the use of a murine model consisting of BALB/c mice for in vivo studies. The program Jalview Version: 2.10.5. was used as a color scale viewer to relate the degree of conservation of the amino acids; additionally, a logo scheme was used to represent the consensus sequence [38]. The possible proteolytic processing sites were estimated through the Site Prediction server http://www.dmbr.ugent.be/prx/bioit2-public/SitePrediction/ (accessed on 17 January 2019) [39] as an indication of the possibility that these proposed molecules could eventually be presented by antigen-presenting cells (APCs) without anticipated compromise of proteolytic degradation in the cellular phagolysosome.

To predict possible linear B epitopes, the ABCPred remote server http://crdd.osdd.net/raghava/abcpred/index.html (accessed on 17 January 2019) was used [40]. The analysis was complemented using the Immune Epitope Database (IEDB) and Analysis Resource, http://www.iedb.org/ (accessed on 17 January 2019) [41,42] and through the BCPREDS servers, http://ailab.ist.psu.edu/bcpred/ (accessed on 20 February 2019) [43,44], and the IEDB Analysis Resource BepiPred-2.0 http://tools.iedb.org/bcell/ (accessed on 20 February 2019) [45]. The cut-off values used were those predetermined by the servers used (default threshold of 0.35 and specificity 75%). The prediction of the possible T epitopes was carried out considering alleles associated with protection against malaria in the framework of the presentation by the major histocompatibility complex II (MHC-II), such as DRB1*01:01, DRB1*13:02, DRB3*03:01, HLA-DQA10101-DQB10501, and HLA-DQA10102-DQB10501 (25,26), through the remote NetMHCII 2.3 Server, available at: http://www.cbs.dtu.dk/services/NetMHCII/ (accessed on 20 February 2019) [46].

### 4.2. Solid-Phase Synthesis of Modified-Antigens and Physicochemical Characterization

Fifteen epitope sequences were obtained by solid-phase synthesis by the 9-fluorenylmethoxycarbonyl (Fmoc) strategy as monomers and polymers formed for a total of 30 polypeptides. Solvents and soluble reagents were removed by filtration. Washings between deprotection, couplings, and subsequent deprotection steps were carried out with *N*,*N*-dimethylformamide (DMF) (5 × 1 min), dichloromethane (DCM) (4 × 1 min), isopropyl alcohol (IPA) (2 × 1 min), and DCM (2 × 1 min) using 1.5 mLof solvent/50 mg of resin each time. The Fmoc group was removed from the resin by two treatments of 15 min with piperidine-DMF (25:75 *v*/*v*). Couplings were performed at 20 °C and monitored using standard Kaiser tests for solid-phase synthesis. For the synthesis of monomer forms, after Fmoc removal of the commercially available Rink amide resin (50 mg, 0.46 mmol/g), the first Fmoc amino acid (0.115 mmol, 5.0 equivalents) was added with 1-hydroxybenzotriazole (HOBt) (18.2 mg, 0.115 mmol; 5.0 equivalents) and *N*,*N*’-dicyclohexylcarbodiimide (DCC) (23.7 mg; 5.0 equivalents) as coupling reagents dissolved in DMF/DCM (7:3, *v*/*v*), and the coupling reaction was stirred for 2 h. Next, the Fmoc group was removed, and a second Fmoc amino acid was incorporated in the resin using the same conditions. The Fmoc removal and the coupling reactions of the rest of the Fmoc amino acids were carried out under the same conditions using 5 equivalents/coupling. Finally, the monomer peptide was Fmoc-deprotected and cleaved from the resin by treatment with a mixture of trifluoroacetic acid-water-triisopropylsilane (TFA/H_2_O/TIS) (95.0:2.5:2.5) for 6 h followed by filtration and precipitation with cold diethyl ether (Et_2_O). Crude products were then triturated 3 times with cold Et2O, dissolved in water-acetonitrile (H_2_O: CH_3_CN) (9:1 *v*/*v*), and then lyophilized. Synthesis of polymer forms was carried out on 150 mg of Rink-amide resin. To further obtain a molecule of high molecular weight represented by a polymer, an active cysteine residue was incorporated at both the *N* and *C* sequence ends. The synthesis of polymer forms was carried out under the same strategy and conditions used for their corresponding monomers (5 equivalents/coupling). Synthesized Cys-peptides were Fmoc-deprotected and cleaved from the resin by treatment with a cleavage mixture including ethanedithiol (EDT) in the system TFA/H_2_O/TIS/EDT (94.0:2.5:1.0:2.5) for 6 h followed by filtration and precipitation with cold Et2O. These crude products were then triturated 3 times with cold Et2O and dissolved in H_2_O:CH_3_CN (9:1 *v*/*v*) and lyophilized as before. Finally, cysteinyl peptides were submitted to disulfide bridge oxidation to obtain the target polymeric molecular forms. Oxidation was carried out from a peptide solution in water (4 mg/mL, pH 7.0) by an oxygen stream under stirring for 16 h. Polymer peptides obtained were dialyzed in water for 24 h using a 500 Da cellulose acetate membrane and further lyophilized as previously published [47].

### 4.3. Peptide Characterization

All *Plasmodium*-based peptide antigen monomer and polymer forms were characterized by analytical reverse-phase-high-performance liquid chromatography (RP-HPLC) and analyzed by matrix-assisted laser desorption/ionization-time-of-flight (MALDI-TOF) mass spectrometry. Analytical RP-HPLC was performed using an Agilent 1200 series chromatography system (Agilent Technologies, Inc., Santa Clara, CA, USA). Analyses were performed on a Zorbax^®^ HPLC C18 column (4.6 µ 50 mm, 5 µm) (Merck KGaA, Darmstadt, Germany), 1 mL/min rate flow, mobile phase system was A:H_2_O/TFA (99.9:0.1 *v*/*v*) and B:CH_3_CN/TFA (99.9:0.06 *v*/*v*), on a 5% to 95% of B linear gradient during 5 min at 25 °C and the UV detector was adjusted to 220 nm. MALDI-TOF mass spectrometry was carried out to the external service to identify a molecular ion of each peptide. Fmoc-Rink Amide MBHA resin and Fmoc amino acids were purchased from Iris Biotech GmbH (Marktredwitz, Germany); DCC and HOBt from AAPPTec (Louisville, KY, USA); piperidine, EDT, TIS, and TFA were purchased from Sigma-Aldrich (Steinheim, Germany); and Et_2_O, DMF, DCM, IPA, and CH_3_CN from Merck KGaA (Darmstadt, Germany). All commercially available reagents and solvents were used as received without further purification. Distilled and deionized water was used for the preparation of all solutions and chromatography eluents [47,48].

### 4.4. Serological Study and Volunteers

Human sera samples *(n* = 252) were obtained from people from malaria-endemic areas in Colombia. Selected areas were Tierralta in the Córdoba department (latitude: 8.167; longitude: −76.067 8°10′1″ north, 76°4′1″ west at 49 m over the sea level); *n* = 72; Tumaco city, Nariño department (latitude: 1.8; longitude: −78.75 1°48′0″ north, 78°45′0″ west at 3 m over the sea level); *n* = 8; Quibdó city in the Chocó department (latitude: 5.683; longitude: −76.65 5°40′59″ north, 76°39′0″ west at 43 m over the sea level), *n* = 67; and San Juan de Nepomuceno in Bolívar department (latitude: 9.95; longitude: −75.0833 9°57′0″ north, 75° 4′60″ west at 156 m over the sea level), *n* = 91. Samples of individuals from non-endemic areas (Bogota DC capital city; north latitude 4°35′56″57 west longitude of Greenwich 74°04′51″30, at 2630 m over the sea level), *n* = 13 were used as an experimental control. Samples were collected considering some inclusion criteria: (i) Volunteers of all ages with a positive diagnosis of *Plasmodium falciparum* infection were selected, without gender discrimination, supported by clinical history, ensuring that they were free of HIV, hepatitis B, and hepatitis C. (ii) As control sera, samples were used from permanent inhabitants of the city of Bogotá (a non-endemic malaria zone of Colombia), without a history of malaria episodes and who had never resided in malaria-endemic areas in the country. (iii) Selected patients had a 10 mL venous blood sample taken in a vacuum tube with EDTA as an anticoagulant, thus plasma samples were obtained. All blood samples were transported to the laboratory according to the WHO Guide on the regulation of transport of infectious substances [49], to be subsequently processed and finally stored at −20 °C until their respective use. (iv) In all cases, sample collection was performed with an informed consent agreement. Data provided by all participants of this study were handled with confidentiality.

### 4.5. Functional In Vivo Activity of Proposed Epitopes

#### 4.5.1. Mice Immunization

Female BALB/c mice (4–6 weeks) were immunized intraperitoneally using 50 µg of each MSP1 native epitope-peptide, as well as with their derived peptide analogs, then were boosted three times with the same dose at days 14, 21, 28, and 35, then formulated and emulsified in Freund’s Complete Adjuvant for the first administration and Freund’s Incomplete Adjuvant for the three boosts [47,48]. Mice in the control group received either saline solution or phosphate-buffered solution—PBS formulated in the same adjuvant. Serum samples were collected previously to the first dose and eight days after each immunization. Four mice were used per group.

#### 4.5.2. Immunoreactivity of MSP1 Peptide Analogs

Enzyme-linked immunosorbent assays (ELISA) were performed to determine the antigenicity of each molecule by assaying anti-peptide antibodies on mouse serum samples. Flat bottom 96-well microplates (Microtest IIITM Falcon F.A.S.T Cat: 3933, Becton Dickinson, Oxnard, CA, USA) were coated with 10 µg/mL of either peptide or its analogs diluted in a pH 9.6 carbonate-bicarbonates buffer and then blocked with 2.5% (*w*/*v*) low-fat milk in PBS-Tween-20 0.05% (*v*/*v*) [44]. Then, 100 µL of each mouse serum (diluted 1:100) was tested in duplicate. To detect stable antigen–antibody complexes, an anti-mouse IgG-peroxidase-conjugate (Vector Laboratories, Catalogue: PI-2000, 1:5000 dilution) was added to the reaction and kept for 1 h at 37 °C. Then, 50 µL of a mixture of H_2_O_2_/Tetramethyl benzidine (TMB-Thermo Scientific™, Waltham, MA, USA) was served as the substrate. A blue color revealed positive reacting mouse antibodies upon hydrogen peroxide degradation as the consequence of the enzyme activity. The reaction was stopped by adding 50 µL of 1M HCl. A Muktiscan microplate reader (Thermo Scientific™, Waltham, MA, USA) was then employed to read Optical < Densities (OD) at 405 nm.

In parallel assays, membrane lysates of *P. falciparum* 3D7 and FCB2 strains were dissolved in Laemmli’s buffer under reducing conditions and resolved by sodium dodecyl sulfate-poly-acrylamide (SDS-PAGE) 10% gel electrophoresis and transferred onto nitrocellulose paper as described in [13]. Pre-immune and immune sera derived from immunized mice were tested in Western blot analysis at a 1:100 dilution. The blot was developed using anti-mouse IgG alkaline phosphatase conjugate (Vector^®^, SK-5400 BCIP/NBT Substrate Kit, 1:10,000 dilution Newark, CA, USA) in a blocking solution. Human sera samples were treated similarly to those obtained from vaccinated mice regarding immunochemical experiments. Hence, the reactivity of antibodies from human sera was established using the same immunochemical techniques (ELISA and Western blotting) but using a peroxidase-anti-IgG human conjugate to detect the formation of antigen–antibody complexes. In this case, control sera samples from permanent inhabitants of non-endemic areas were used as a negative control. Cutoff points for ELISA were calculated as three SD above the mean OD value at 450 nm of a normal mouse serum sample or a serum sample from healthy volunteers who had never been exposed to malaria.

#### 4.5.3. Vaccination of BALB/c Mice and Experimental Challenge with Rodent Malaria Species

All groups of immunized BALB/c mice were intraperitoneally (i.p) inoculated with a lethal dose of 5 × 10^4^ infected red blood cells (iRBCs) with *P. berghei* ANKA and *P. yoelii* 17XL in simultaneous experiments by using malaria-infected inoculum from two donor mice having parasitemia levels higher than 50% as a described in [23,50].

Parasitemia levels were followed by using Giemsa staining of fine blood smears and examined at 1000× magnification. Parasite-infected red blood cell (iRBC) percentages were analyzed by recounting at least 9000 RBCs to establish the evolution of parasitemia in all animal groups. Animal care was carried out in agreement with international guidelines [51]. Animals subjected to invasive procedures were treated under anesthetic or analgesic conditions as ethically recommended.

#### 4.5.4. CellularTh1/Th2 Immune Responses of Malaria-Protected Animals

After 20 days of being experimentally challenged with lethal doses of *P. berghei ANKA* and *P. yoelii 17XL*, all immunized and control mice groups were sacrificed by deep anesthesia and cervical dislocation, following international guidelines for the use of experimental animals as mentioned above [51].

To determine the cytokine expression profiles in those animals who efficiently controlled and survived the malaria challenge, the spleen was aseptically removed, and spleen cells were harvested and frozen at −120 °C until use. For the cytokine detection experiment, cell suspensions from each treatment were thawed, reconstituted, and carefully washed three times with serum-free RPMI 1640 medium, then cultured in flat-bottom 96-well plates (Thermo Scientific™ Nunc™ MicroWell™) and centrifuged at 1000 rpm for 5 min, and then suspended at a concentration of 5 × 10^6^ cells per well in complete RPMI 1640 medium supplemented with 20% fetal bovine serum (Hyclone, GE, IL, USA).

Cells of each well were stimulated with the inducer site-directed designed peptidomimetic at 200 nM concentration and incubated at 37 °C and 5% CO_2_ in a humidified chamber. Supernatants of each cell culture were harvested after 48 h and 72 h of antigen stimulation. A 25 µg/mL solution of phytohemagglutinin (PHA) (Thermo Fisher Scientific-Invitrogen, Waltham, MA, USA) was used as the positive control. The 96-well plates were then incubated with dilutions of anti-cytokine antibodies to Interleukin-2 (IL2), Interleukin-4 (IL-4), Interleukin-5 (IL-5), Interferon-γ (INFγ), and Tumor necrosis factor (TNF) (BD™ Cytometric Bead Array, BD Biosciences, CBA Mouse Th1/Th2 Cytokine Kit, Catalog No. 551287, San Diego, CA, USA) in accordance with the manufacturer’s protocol. Then, a developer solution was added, and plates were submitted for reading on a Flow cytometer FACSCanto II (Becton Dickinson BD Biosciences, San Diego, CA, USA). Results were expressed as the Stimulation Index (SI), calculated by dividing the results of the stimulated number of cells by each peptidomimetic antigen regarding the number of cells without stimulation and data were processed with Graph Pad Prism 8.0.2 software according to protocols published elsewhere [52].

#### 4.5.5. In Silico Docking Experiments, Getting-Ready Protocols

All molecules were submitted to a getting-ready protocol conducted in a computation system provided by Linux Ubuntu versión 11.0-2022. Input information used was obtained from two sources; the protein database Data Bank was used for all PDB format files for protein coordinates data. Files for malaria-resistant and susceptible class-II HLA molecules were selected and downloaded. Thus, PDB files of HLA-DRB10101 code 1aqd and HLA-DRB10401 code 5jlz were obtained. Additionally, a T-cell receptor TCR coordinate file was also obtained, whose code is 1j8h. Additionally, the native non-modified peptide MSP1 code 2mu7, as well as the PDB of two peptidomimetics analog files named MSP1-V^52^ψ[CH_2_-NH]-L^53^- (B1 analog 2) and MSP1-M^51^ψ[CH_2_-NH]-V^52^- (B1 analog 1), were employed. After obtaining and downloading the PDB files, they were adjusted for all the target epitope-peptides, their modified analogs, HLA-II molecules, and a human TCR 3D structure. For all files, a minimization energy step was initially carried out to adjust the orientation of the side chains, adding missing hydrogens and assigning formal charges as required at a pH of 7.0. This getting-ready protocol was performed with Maestro software (Maestro, Schrödinger Schrodinger.com; Schrödinger 2022-2: Maestro, Schrödinger, LLC, New York, NY, USA, 2021), a software under an academic license agreement. Additionally, as preparation before the docking calculation, only one molecular dynamics calculation of 2 ns was carried out for all peptide molecules to relax the structure and guide the most stable conformation, at 310.15 K with a 0.015 M KCl concentration. The molecular dynamics calculation was performed with the Gromacs software version for Linux adjusted with standard parameters, as indicated by the supplier. Gromacs is free software, available under the GNU Lesser General Public License (LGPL), version 2.1. To finish the preparation of the input files, molecular dynamics calculation steps were performed with Gromacs (a Charmm-Gui.com web server tool). Thus, coordinate files having all desired parameters were generated to formally start the calculation once a molecular dynamics step was performed. These files were processed again to remove all water molecules and KCl ions, therefore free ions and water molecule structures were generated, mainly to avoid interferences in the docking calculation step. The structure adjustment was carried out with the Pymol by Schrödinger (Pymol pymol.org; Pymol Molecular Graphics System, version 2.0 Schrödinger, LLC. Cambridge, MA, USA) software under an academic license.

The molecular coupling study was carried out under the ZDock online platform, under an academic license (ZDock Server: references (Umassmed.edu (accessed on 15 August 2022)). Files obtained in the preparation steps and after molecular dynamics were directly loaded on the platform in three stages. Initially, the file corresponding to the receptor is loaded, and subsequently the file corresponding to the peptide (whether native or its modified analogs) is loaded, and finally a third file “.txt” is loaded where the active or involved residues are indicated in the molecular recognition receptor peptide, to indicate to the docking algorithm which amino acids are involved in possible binding from both the receptor and the peptide, simplifying the processing time. At the end of the time required for the docking, a file is generated and downloaded and represents the 1000 best structures for possible dockings, where the shape and orientation of the peptide in the cleft or active sites of the receptor may vary. This file is used as input for a structure refinement program, Fiberdock version 3.0 (Fiberdock Home, Tau.ac.il), which refines the peptide adjustment in the HLA-binding active pockets through movement and guidance of the side chains and flexibility the peptide backbone, taking as reference the 1000 models proposed by the ZDock algorithm.

#### 4.5.6. Selection of Structures’ Ternary Complexes and Docking Models

After the refinement process of the molecular docking by Fiberdock, the platform generates the 10 best structure clusters for each peptide analog studied, selected from the lowest free Gibbs energies for each structure, the first structure is the one that has the lowest energy, and so on increasing to construct 10. For the final selection of the complex that would be selected to be used for the comparison and analysis, two criteria were established to define the study complex of the ten predicted for each molecular pair (receptor-peptide), being i—the theoretical and spatial alignment of the native peptide or the one modified with the active cleft (or active site) of the HLA-II complex; ii—the receptor–peptide complex with lesser free energy that is available within the complexes that are better spatially aligned.

After the selection of the best complexes, a new docking calculation was carried out to establish the interaction of the HLA-peptide complex generated previously with the TCR, and thus obtain the interaction of HLA-peptide/analog-TCR. In the same way that the docking calculation was carried out in the first step, besides all molecular refinements, minimization dynamics, and selection of the best complexes, it was carried out similarly for this second stage, where it was intended to determine the trimolecular interaction between HLA molecules, the native or modified peptide, and the T cell receptor. Ending this stage, it is expected that a “.pdb” file is obtained that is selected based on the best selection criteria, thus having the lowest possible free Gibbs energy within the possible docked proposed solutions. A final molecular dynamics calculation on the representative structures for each of the performed dockings, that is, for each peptide, native and modified, together with the two coupled receptors (HLA + peptide + TCR) is carried out taking into account the following parameters and settings: i—temperature 310.15 K, ii—Charmm36m force field, iii—0.015 M KCl concentration, iv—standard and adjusted to the size of each complex for the case size, and v—constant pressure at 1 atm. Three-dimensional structure visualization was also performed by Maestro of Schrödinger.

#### 4.5.7. Molecular Dynamics Simulations of Highly Immunogenic MSP1 Peptide Analogs

Simulations for the single MSP1^42–61^ fragment and its modified analogs on free conformational stages were performed using the GROMACS molecular dynamics package1, version 2022.4 (released on 16 November 2022). The OPLS-AA force field2 was employed for all the peptides and force field parameters were obtained through the ffld_server module of the Schrodinger software3 and ffconv.py script4 was used to convert into Gromacs topology format. The TIP3P water model 5 was used for solvating all peptides. Native peptide systems contain 3216 water molecules, MSP-V^52^-ψ[CH_2_NH]-L^53^ (**B1 analog 1**) peptide systems contain 2101 water molecules, and MSP-M^51^-ψ[CH_2_NH]-V^52^ (**B1 analog 2**) peptide systems contain 3262 water molecules. All systems were energy-minimized using 5000 steepest descent 6 steps. The systems were then equilibrated for 100 ps using the canonical (NVT) ensemble, followed by a further 100 ps of equilibration simulation with the isobaric-isothermic (NPT) ensemble, and during equilibration restraint were employed on the peptides. The production simulations for all the systems were performed using an NPT ensemble without any restraint on the peptides. Water molecule bond length was constrained with the SETTLE algorithm 7 and long-range electrostatic interactions were calculated using the particle mesh Ewald8 (PME method with a cutoff of 10 Å) and van der Waals (vdW) interactions were calculated using a cut-off of 10 Å. All MD simulations were performed at a temperature of 310.15 K and a pressure of 1 bar. For temperature coupling, Velocity-rescale algorithm 9 was employed and for pressure coupling Parrinello-Rahman method 10 was used. A leap-frog algorithm was used for integrating Newton’s equations of motion with an integration time of 2 femtoseconds (fs). For each peptide, two simulations were performed using different velocities, randomly generated by the Gromacs simulation package. Each simulation was performed for 200 nanoseconds (ns). So, we have six trajectories and an overall simulation time of 1.2 µs. All analysis was performed using the Gromacs simulations package.

## 5. Conclusions

Evidence from this work demonstrated that site-directed modifications introduced into selected epitope-peptides derived from the MSP1 *Plasmodium* spp. antigen direct a given immune response against a controlled malarial experimental infection of a rodent animal model. This suggests that the design of peptidomimetics and the evaluation of their functional activity through available murine malaria models are suitable for identifying promising sequences as possible components of immunoprophylactic formulations against malaria, such as **B1An4**, **B1An2**, **T3An5**, **T3An4**, **T3An8**, **B6An1**, and **B6An2**.

Proposed peptidomimetics from this research were recognized at a high reactivity degree when tested against naturally induced human antibodies and were able to stimulate significant immunogenicity in mice. Animals selectively vaccinated with designed epitope-peptides were able to control parasitemia, resolved infection in some cases, and stimulated Th1 lymphocytes, showing remarkable antigenic and immunological functional properties; thus, some of them could be included in the formulation of a vaccine sub-unit against malaria.

Strengthened insights into the analysis of MSP1 protein-derived structurally modulated antigens, as one of the most exciting malaria targets, could be regarded as an essential contribution to the generation of new knowledge on Plasmodium biology at a molecular level, therefore contributing to the understanding of the still unknown immune response mechanisms to this devastating microorganism. A direct consequence of this knowledge would be its application for developing alternative tools for preventing emerging pandemics and outbreaks caused by viral pathogens [53].

## Figures and Tables

**Figure 1 molecules-28-02527-f001:**
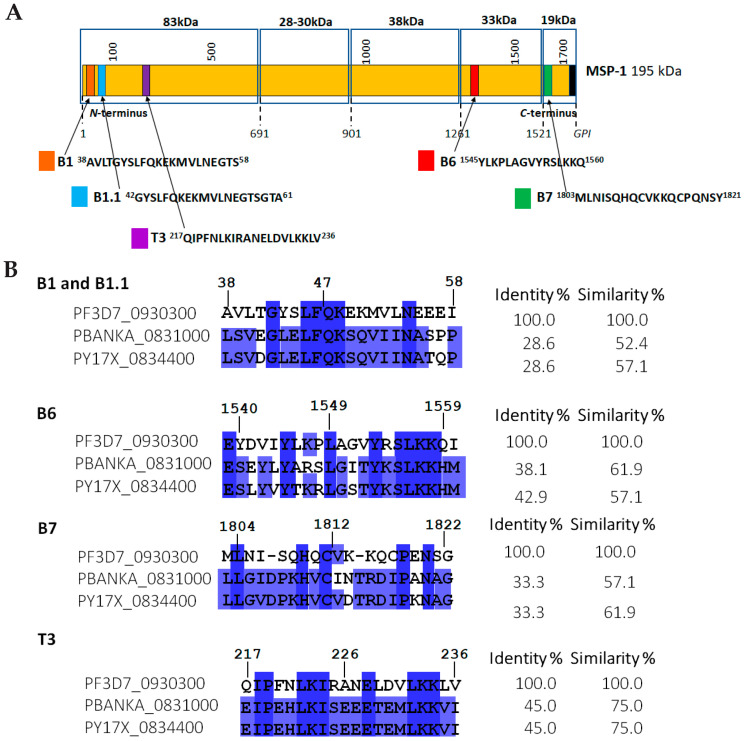
Epitope-peptide selection from the MSP1 structure. (**A**) Four B and one T potential epitopes are located at *N-* and *C-*MSP1 termini herein coded **B1**, **B1.1**, **B6**, **B7**, and **T3**. (**B**) Structure homology studies show the high identity and similarity of the MSP1 fragments among human and rodent *Plasmodium* spp. Color code for sequence identity: dark blue higher than 80%, mid blue higher than 60%, light blue higher than 40%, and white for no identity.

**Figure 2 molecules-28-02527-f002:**
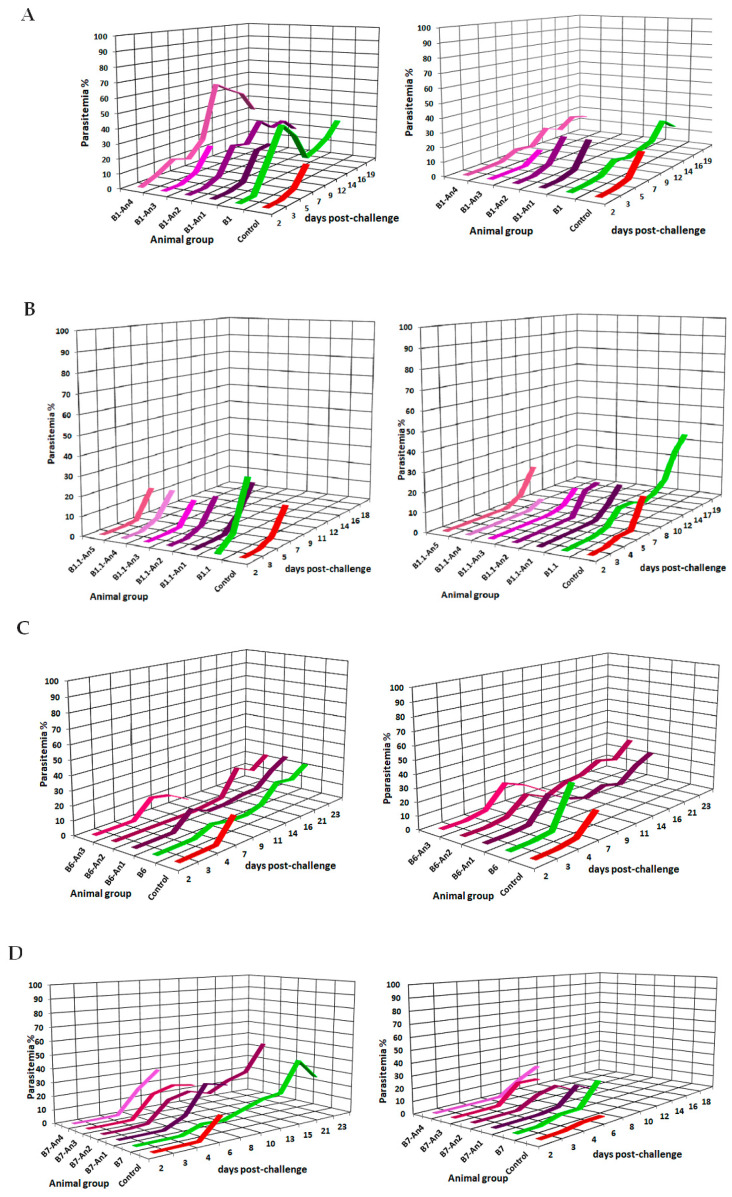
Effect of B and T MSP1 selected epitope-peptides on rodent malaria infection. The left panels show the effect of each B- and T-designed epitope-peptides regarding *P. berghei* ANKA infection levels while the right panel displays their effect on *P. yoelii* 17XL parasitemia levels. Color codes are red for control groups administered with isotonic saline solution, green for groups vaccinated with the non-modified sequence, and purple–dark to light scale for their sequence analogs. (**A**) **B1** epitope-peptides, (**B**) **B1.1** epitope-family members, (**C**) **B6** epitope-analog family, (**D**) **B7** epitope-member family, (**E**) **T3** epitope-analog peptidomimetics.

**Figure 3 molecules-28-02527-f003:**
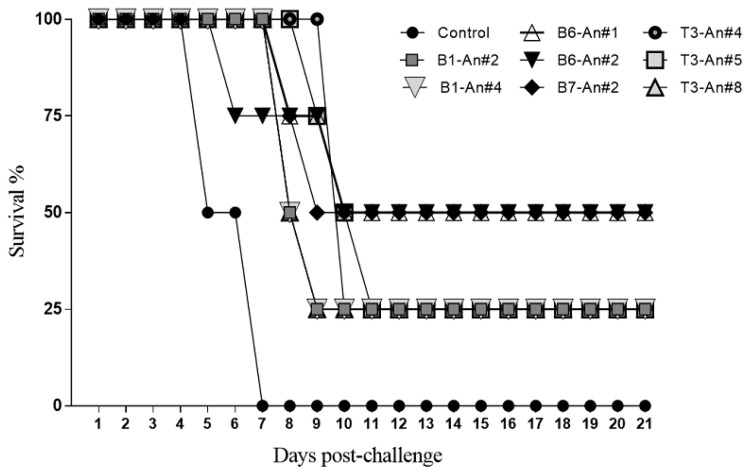
Survival profiles of BALB/c mice immunized with MSP1 epitope-peptidomimetics. Animals were vaccinated with native and modified potential MSP1 B- and T-epitopes, in a four-dose immunization scheme and subsequently were challenged by receiving 5 × 10^4^ iRBCs with *P. berghei* ANKA and *P. yoelii* 17XL strains. As observed, some animals controlled parasitemia levels and some of them survived even after the observation time of 20 days post-infection, and the control group died in the first 7 days after being challenged. In all cases, animal groups had four individuals which were split before simultaneous experimental challenging. The control groups were administered with isotonic saline solution.

**Figure 4 molecules-28-02527-f004:**
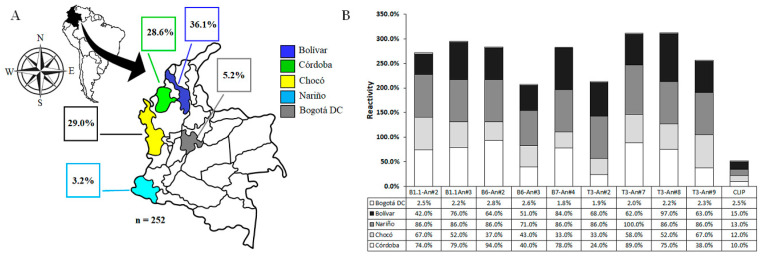
Selected provinces of malaria-endemic and non-endemic areas in Colombia. (**A**) A total of 252 human sera samples were collected and proportions are shown in agreement with their department (province) of origin. (**B**) Recognition distribution of selected MSP1 epitope-peptides by human sera antibodies by ELISA tests. CLIP (class-II-associated invariant chain peptide).

**Figure 5 molecules-28-02527-f005:**
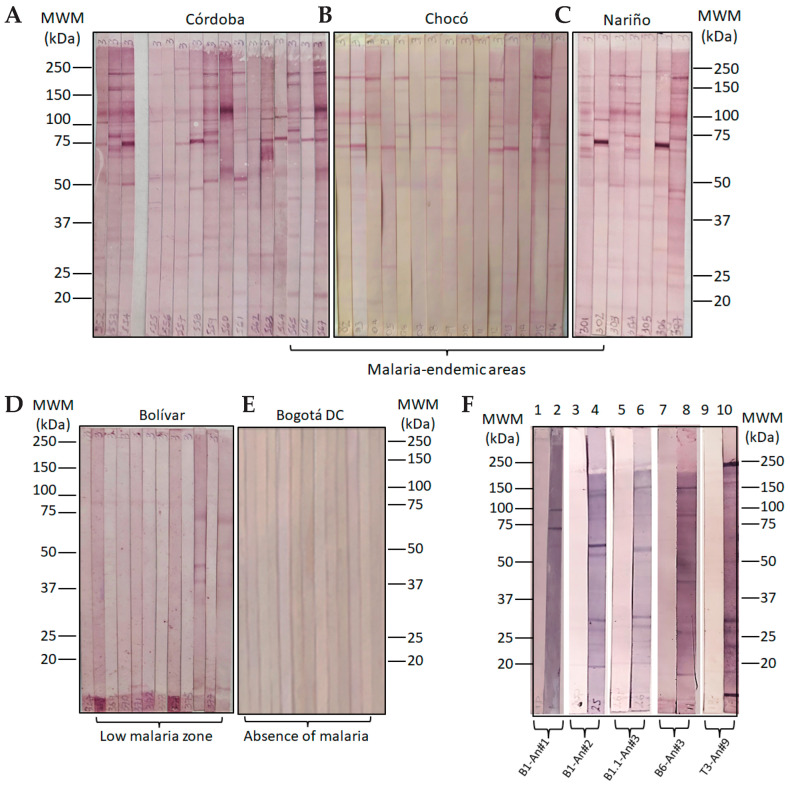
Western blot analysis for reactivity of human antibodies to surface proteins of *P. falciparum* 3D7, and comparison with vaccinated BALB/c mice with MSP1 site-directed epitope-peptides. Reactivity of human sera from malaria-endemic areas is shown in (**A**–**E**) for the control group and (**F**) for MSP1 epitope-peptide vaccinated mice sera reactivity.

**Figure 6 molecules-28-02527-f006:**
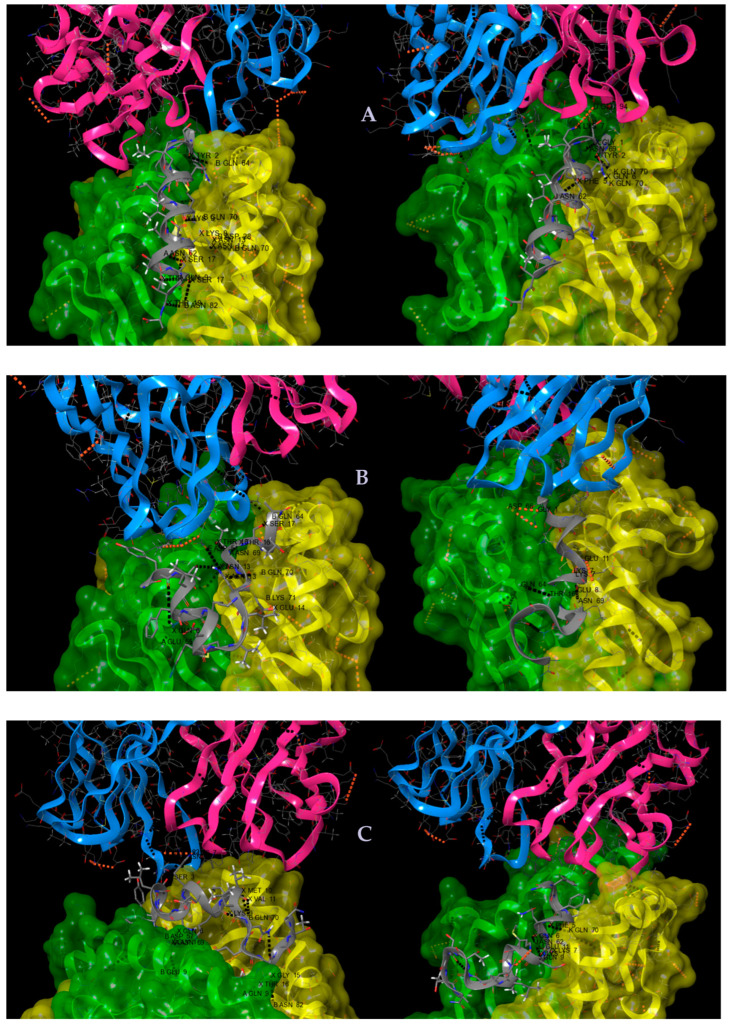
Stability of HLA-designed epitope-TCR ternary complexes. The malaria-sensitive associate HLADRB1*04:01 allele complexed to the B.1.1 (MSP^142–61^) molecule in (**A**); two modified analogs B1.1 analog 2 (MSP1-V^52^-ψ[CH_2_-NH]-L^53^-) in (**B**); and the B1.1 analog 1 (MSP1-M^51^-ψ[CH_2_-NH]-V^52^-) is shown in (**C**). Each is molecularly docked to the vα, β TCR chains forming some ternary complexes stabilized by H bonds (black dotted lines) and salt bridges (red dotted lines). Colors for HLA α-chains are seen in green, β-chains in yellow, vα-TCR chains in salmon pink, and vβ-TCR chains in pale blue.

**Table 1 molecules-28-02527-t001:** Characteristics of the MSP1 peptidomimetics.

	Epitope-Peptide	Modification Position	Antigenicity ^a^	Anti-Molecule Ig Titers
**B1**	B1	*Pf.3D7.*MSP-1^42–61^	**	1:400
	B1-An1	*Pf.3D7.*MSP-1-G^42^*-*ψ[CH_2_-NH]-Y^43^-	**	1:800
	B1-An2	*Pf.3D7.*MSP-1-Y^43^*-*ψ[CH_2_-NH]-S^44^-	**	1:51,200
	B1-An3	*Pf.3D7.*MSP-1-L^45^*-*ψ[CH_2_-NH]-F^46^-	NS	ND
	B1-An4	*Pf.3D7.*MSP-1-M^51^*-*ψ[CH_2_-NH]-V^52^-	****	1:400
**B1.1**	B1.1	*Pf.3D7.*MSP-1^38–58^	**	1:400
	B1.1-An1	*Pf.3D7.*MSP-1-V^52^*-*ψ[CH_2_-NH]-L^53^-	*	1:200
	B1.1-An2	*Pf.3D7.*MSP-1-M^51^*-*ψ[CH_2_-NH]-V^52^-	**	1:200
	B1.1-An3	*Pf.3D7.*MSP-1-K^50^*-*ψ[CH_2_-NH]-M^51^-	**	1:12,800
	B1.1-An4	*Pf.3D7.*MSP-1-E^49^*-*ψ[CH_2_-NH]-K^50^-	***	1:200
	B1.1-An5	*Pf.3D7.*MSP-1-K^48^*-*ψ[CH_2_-NH]-E^49^-	**	1:400
**B6**	B6	*Pf*.3D7.MSP-1^1545-1560^	**	1:200
	B6-An1	*Pf*.3D7.MSP-1-Y-*d*L^1546^-K-	**	1:6400
	B6-An2	*Pf*.3D7.MSP-1-Y-*d*K^1547^-P-	**	1:800
	B6-An3	*Pf*.3D7.MSP-1-K-*d*P^1548^-L-	**	1:200
**B7**	B7	*Pf.3D7.*MSP-1^1803*–*1821^	**	1:400
	B7-An1	*Pf.3D7.*MSP-1-M^1803^*-*ψ[CH_2_-NH]-L^1804^-	**	1:400
	B7-An2	*Pf.3D7.*MSP-1-L^1804^*-*ψ[CH_2_-NH]-N^1805^-	****	1:400
	B7-An3	*Pf.3D7.*MSP-1-K^1815^*-*ψ[CH_2_-NH]-Q^1816^-	*	1:200
	B7-An4	*Pf.3D7.*MSP-1-C^1817^*-*ψ[CH_2_-NH]-P^1818^-	**	1:200
**T3**	T3	*Pf.*3D7.MSP-1^217–236^	***	1:200
	T3-An1	*Pf.3D7.*MSP-1-L^222^*-*ψ[CH_2_-NH]-K^223^-	****	1:1600
	T3-An2	*Pf.3D7.*MSP-1-K^223^*-*ψ[CH_2_-NH]-I^224^-	NS	ND
	T3-An3	*Pf.3D7.*MSP-1-R^225^*-*ψ[CH_2_-NH]-A^226^-	***	1:1600
	T3-An4	*Pf.3D7.*MSP-1-N^227^*-*ψ[CH_2_-NH]-E^228^-	*	1:3200
	T3-An5	*Pf.3D7.*MSP-1-L^229^*-*ψ[CH_2_-NH]-D^230^-	*	1:1600
	T3-An6	*Pf.3D7.*MSP-1-D^230^*-*ψ[CH_2_-NH]-V^231^-	**	1:800
	T3-An7	*Pf.3D7.*MSP-1-V^231^*-*ψ[CH_2_-NH]-L^232^-	****	1:51,200
	T3-An8	*Pf.3D7.*MSP-1-L^232^*-*ψ[CH_2_-NH]-K^233^-	****	1:51,200
	T3-An9	*Pf.3D7.*MSP-1-K^234^*-*ψ[CH_2_-NH]-L^235^-	**	1:51,200
	T3-An10	*Pf.3D7.*MSP-1-L^235^*-*ψ[CH_2_-NH]-V^236^-	**	1:3200

^a^ Antigenicity: Data processed in Graph Pad Prism 8.0.2. Paired analysis of all the readings by group Pre-I Vs Post 4th (*p* < 0.05 *; *p* < 0.01 **; *p* < 0.001 ***; *p* < 0.0001 ****; NS: not significant). Antibody titers were defined as the sera’s highest dilution that still yields a positive reading expressed as the OD mean value at 450 nm produced by a given analyte sample—3SD: the OD mean value of pre-immune sera ± SD; ND: not detected.

**Table 2 molecules-28-02527-t002:** Th1/Th2 profile of spleen cells from vaccinated mice.

	Molecule	TNF	IFNγ	IL-2	IL-4	IL-5	Associated Immune Pattern
		[pg/mL]	[pg/mL]	[pg/mL]	[pg/mL]	[pg/mL]	
	**PHA**	6.35	139.86	366.31	17.98	221.93	
	**Control**	3.81	1.18	17.01	4.73	ND	
**Cell stimulation at 48 h**	**B1**	15.16	2.32	202.04	9.29	33.86	**Th1**
	**B1An4**	ND	1.12	ND	3.31	ND	**Th1/Th2**
	**B1An2**	3.59	1.22	79.29	8.08	1.05	**Th1**
	**B1.1**	25.07	1.25	108.78	5.65	3.25	**Th1**
	**B6An1**	26.11	1.58	110.59	2.87	6.54	**Th1**
	**B6An2**	7.75	1.28	7.17	5.11	11.12	**Th1/Th2**
	**T3**	14.82	1.35	94.90	4.64	ND	**Th1**
	**T3An5**	ND	1.16	10.09	6.66	3.54	**Th1/Th2**
	**Control**	3.61	1.06	ND	ND	ND	
**Cell stimulation at 72 h**	**B1**	22.44	1.39	38.44	3.86	2.68	**Th1**
	**B1An4**	ND	1.05	ND	3.73	ND	**Th1/Th2**
	**B1An2**	ND	1.26	118.65	6.01	3.41	**Th1**
	**B1.1**	9.97	1.14	15.6	6.07	6.34	**Th1/Th2**
	**B6An1**	97.25	2.38	324.1	13.37	21.99	**Th1**
	**B6An2**	47.22	1.46	43.36	8.06	9.36	**Th1**
	**T3**	6.18	1.29	74.65	4.72	3.32	**Th1**
	**T3An5**	ND	1.12	12.34	5.21	5.64	**Th1/Th2**

ND for non-determined, PHA phytohemagglutinin, and Control is spleen cells from a normal mouse neither immunized nor stimulated with any antigen.

## Data Availability

Not applicable.

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
