# Peer review of "Natural Plasmodium falciparum Infection Stimulates Human Antibodies to MSP1 Epitopes Identified in Mice Infection Models upon Non-Natural Modified Peptidomimetic Vaccination"

_molecules, 2023, doi:10.3390/molecules28062527_

Round 1

Reviewer 1 Report

  Natural Plasmodium falciparum infection stimulates human antibodies to MSP1 epitopes identified in mice infection models 3 upon non-natural modified peptidomimetic vaccination by Rodrigues et al.

With the aim to contribute to malaria vaccines in P. falciparum, the manuscript comprises the identification of peptides from MSP1 N- and C- terminal and the capacity of native and modified ones in inducing B and T cell responses in mice, which thereafter challenged with rodent malaria and follow the parasitemia levels. Antibodies IgG against peptides and a “Pf-surface proteins” were measured by ELISA in samples from humans infected with P. falciparum or immunized mice. Natural and modified peptides were synthetized and tested its activity to induce antibody and T cell immune responses in immunized mice

The manuscript is interesting, however, in its current version it is tedious to read. The manuscript requires thorough revision and proper organization of the sections before is considered for publication. The results also include portions of methods and discussion, which makes difficult to follow, and maybe because of that, the description of the results per se is vague and incomplete. As well, some results are confusing, over interpreted or misinterpreted. There also inconsistencies between sections. Authors do not explain why B and T cell peptides were tested its capacity to induce antibodies and T cell responses, means that these epitopes were promiscuous?. The discussion repeats interpretations incorporated in the result section. Major and minor suggestions are listed below mainly on the points that understood best.

Major points

Introduction

There are some previous results that were integrated in the results section, it should be mentioned in the introduction or methods sections. (lines 314-317): “The B1 and B1.1 overlapping sequences have been previously studied for their immunogenic capacity to stimulate human T lymphocytes, as well as because both harbor high-affinity binding motifs to RBCs [17,19]. Previously it has been described the MSP1-316 Y43-ψ[CH2NH]-S44 peptide-bond isostere here in coded B1An2 as a determinant for recognition by T lymphocytes in the context of human HLA-II defined alleles [13].”

Results

Please explained clearly how the peptides were selected as in section 2.1, as did not mention what is mentioned in the conclusions about HLA-II and screening

This section is difficult to follow, and need thorough revision, this is unnecessary large and has a pattern. For each result it is included an introduction, results and discussion sentences and some methodology (e.g. lines 443-461). Please arrange the results and the manuscript as indicated in the authors guidelines; revise and place the information accordingly. Sometimes, before description of the result, the interpretation is mentioned (e.g. lines 447-449). Also, avoid redundant paragraphs e.g. in the results and the methods sections e.g. lines 99-109 vs lines 696-707. Also see below.

These are not results:  (line133-139) “The first experimental step consisted in synthesizing all five MSP1 epitope sequences and a family of non-natural peptide analogs among -ψ-[CH2–NH]- peptide isostere bonds (backbone modifications) and single D-amino acid substitutions (side-chains chirality) by liquid and solid phase Fmoc chemistry and then spectrophotometrically characterized. Each peptide surrogate was strategically positioned as designed (Table 1). “Then, subsequent functional activity of each native and modified sequence was evaluated by immunization of female BALB/c mice groups, as well as by immunochemical tests with human sera from malaria-endemic regions of Colombia.”

These are not results (lines186-189) “It has been described in the literature that mice of the BALB/c strain are susceptible to infections by Plasmodium berghei ANKA and P. yoelii 17XL, which present tropism for mature red blood cells (RBCs) and reticulocytes. The infection is considered to be lethal between 6 to 8 days post-infection and so this mice strain has been regarded as a suitable malaria infection model [16].”

It seems the following paragraph should be in a section other from results (lines 364-369): “Previous work evidenced variations in the antigenic presentation mechanisms when introducing peptide backbone modifications revealing that a single isosteric bond -ψ [CH2-NH]- is capable of modulating conformational, as well as the peptide fragment binding properties to MHC- II, acting as a new category of peptide ligands. Also, the role of a stereochemical modulation on an epitope conformation resulted in shifting of the three dimensional conformation of a given epitope, which seems to be crucial for a differential functional activity for controlling a malaria”

It is confusing, what is the correspondence of having 2% samples from Bolivar with reactivity to Pf-surface proteins vs the reactivity to 9 peptides being 42-97%?.  Whats is (line414) “strongly recognized” means the high antibody titer? Data of ELISA titers or ranges are not shown. The results from Figure 2B seem not to be describe in the text.

The following sentence has problems and it seems not to be part of the result section. Authors might be over interpreting the results, assumed that by measuring total IgG (line838) antibodies also subtypes were measured as others ( lines 417-418) “IgG1 and IgG3 sub-classes have been considered cytophilic and protective against P. falciparum infection [26, 27]. “

and please clarify in the result section what results and methods are related to the sentence on lines 418-423: “Immunoglobulin isotype repertoire present in the analyzed sera samples seems to include high proportions of these Ig subclasses, evidencing that antibodies stimulated by malarial natural infection episodes, and specifically react with the artificially produced epitope-peptides de-signed in this study.”

Samples came from different type of host, human Pf-infected an mice immunized with peptides, the later must have a more specific reactivity ( maybe against the native PfMSP1 protein), if not, please explain.  

It seems a misinterpretation: why mention that immune plasma from Pf-infected humans produce similar reactivity than sera from peptide-vaccinated mice to Pf surface proteins by western blot, was that expected and why? If these are two different types of antigen. Lines430-433. “country had comparable protein-band recognition of vaccinated mice sera as shown in Fig. 5A-E. Sera from the groups of vaccinated mice and human IgG antibodies reacted similarly with Plasmodium falciparum 3D7 and FCB2 strains´ surface proteins present in membrane lysates thereof when tested in Western blot assays, as described in Table 2.” Also mentioned on line 662-664. Moreover, table 2 does not show results from  western blot assays.

Table1. In this table please add the Ig titers to the native peptides for B1, B1.1, B 6 and B7. Please explain, why the presumed T cell epitope peptides (of about 20 amino acids) were tested its ability to induce Ig antibodies? Was expected to have a promiscuous peptides? It is not clear, what result is anti-molecule Ig, what samples were tested? Mice samples? Of what day post immunization?

Table 2  has some problems, if fact it does not show the Th1-Th2 profile itself, this shows the levels of different cytokines. why the control for IL-5 was not determined? Why B6An2 or T3An5 with baseline results for all/most cytokines are indicated an immune pattern Th1-Th2? Please provide information about the criteria to discern the immune pattern/Th1/Th2 profile. It is not shown in the method´s section 4.5.4.

Figure 2. the style of the graphs makes difficult to understand the kinetics c of the parasitaemia learly, if possible change the format.

If immune plasma were obtained from Pf-infected humans should be indicated in section 2.3: (line372) “individuals exposed to natural Plasmodium spp infection “ vs (lines 794-795) “Samples were collected considering some inclusion criteria: i) Volunteers of all ages were selected, with a positive diagnosis of Plasmodium falciparum infection,….”

Figure 4B is not described in the text.

Figure 5. not indicated in the text. It shows Western blots with “surface proteins of P. falciparum” reactivity with plasma samples from Pf infected humans and sera from vaccinated mice with peptides. Plasma from endemic areas reacted with several bands, one of about 200kD not seen in all samples. Lines 383-386 what means the percent ( % ) plasma reactivity producing certain pattern of bands? Or ELISA results? It is not clear.  Figure 5 seems to be misplaced?

Please describe in the manuscript what is?  (line 345) “surface proteins of Plasmodium falciparum FCB2 lysates [21].” Should indicate briefly in the method section, how it was obtained and surface antigen from what stages? Call my attention that these analysis/results seem not to be articulated with the results from mice immunization and challenge. Also, in the conclusions seems not to fit, the high prevalence of antibodies in Pf infected individuals

If mice were boosted 3 times why there are four days (Line 810) “were boosted three times with the same dose at days 14, 21, 28, and 35 formulated”

Serum samples were obtained from mice: preimmune and 8-days after each immunization. All of them were tested for antibodies? Or what samples?

 Lines 836-837: “The reactivity of antibodies from human sera was established using the same immunochemical techniques (ELISA and western blotting),…” It was the ELISA describe above? Plasma samples were tested at 1:100 as the mice ones, please add this information.

Discussion

Need a thorough revision as seems that mostly repeat what was discussed in the results section.

Minor

Revise the entire manuscript as in many occasions genus/species or in vivo ,  ex vivo need to be Italianized e.g. lines 17, 20, 272, 273, 298, 299, 457,…383, 996, 654 etc

Samples from malaria endemic areas were obtained using an anticoagulant (Line 800: “…blood sample taken in a vacuum tube with EDTA… “), then should be mentioned as plasma samples (line 30, and check others).

Several typos should be revised e.g. whit (line82, 85), line 506 “(1j8h). [35].”, line273 “level. C Color”

Figure 1. please indicate the following “light blue higher than 40%, light gray lower than 40% in white for no identity” f and should change to ranges e.g. 60-80% if not, higher than 60 comprises everything from 60 to 100%. Please clarify.

Line349 might mean: C-terminal (19kDa)

Please be consistent, it is sometimes BI.1 An1 and others BI.1 An#1…………

Indicate in the Figure4 legend, what means CLIP?

Add the reference: (line 858) “following international guidelines for the use of experimental animals as mentioned above.”

Reviewer 2 Report

In the submitted manuscript, the authors identified and investigated the antigenicity/immunogenicity of different malaria B- and T -cell epitopes. The study is interesting and well-written and only minor changes are needed. For example, the results of western blot should be clearly explained and cited in the results section. The authors need to ensure that all figures are readable and also check for typos especially the materials and methods section.

Round 2

Reviewer 1 Report

Natural Plasmodium falciparum infection stimulates human antibodies to MSP1 epitopes identified in mice infection models upon non-natural modified peptidomimetic vaccination

Authors made the revision, however there are some minor aspects to revise before it is accepted for publication.

Line 37 of the abstract: please make clear “for this disease detection” means malaria parasite detection?

In the introduction review the last paragraphs because it is not clear if the style or content of the paragraphs is in the right place. e.g. lines 101-107 seem like perspectives that should be at the end of the discussion, or lines 99-100 seem like previous work that has no reference and perhaps should go in methods. Please review the contents, their relevance to this section, and the way they are expressed in lines 92-107.
